# An Experimental Assessment of People’s Location Efficiency Using Low-Energy Communications-Based Movement Tracking

**DOI:** 10.3390/s22229025

**Published:** 2022-11-21

**Authors:** Marius Minea

**Affiliations:** Department of Telematics and Electronics for Transports, Transports Faculty, University Politehnica of Bucharest, 060042 Bucharest, Romania; marius.minea@upb.ro; Tel.: +40-21-402-9653; Fax: +40-21-318-1012

**Keywords:** indoors localization, people’s density and movement tracking, low-energy communications, GPS-denied environments, Bluetooth sensors array, trilateration, received signal strength indicator

## Abstract

(1) Background: public transport demand dynamics represents important information for fleet managers and is also a key factor in making public transport attractive to reduce the environmental footprint of urban traffic. This research presents some experimental results on the assessment of low-energy communication technologies, such as Wi-Fi and Bluetooth, as support for people density and/or movement tracking sensing technologies. (2) Methods: the research is based on field measurements to determine the percentage of discoverable devices carried by people, in relation to the total number of physical persons in interest, different scenarios of mobile devices usage and evaluation of influences on radio signals’ propagation, RSSI / RX read values, and efficiency of indoor localization, or in similar GPS-denied environments. Different situations are investigated, especially public transport-related ones, such as subway stations, indoors of commuting hubs, railway stations and trains. (3) Results: diagrams and experiments are presented, and models of signal behavior are also proposed. (4) Conclusions: recommendations on the efficiency of these non-conventional traveler and passenger flow tracking solutions and models are presented at the end of the paper.

## 1. Introduction

The present research focuses on the efficiency of non-conventional detection of travelers’ presence and flow, based on discoverable portable devices such as smartphones, smart watches, BT headphones, etc. The purpose is to use this information in the evaluation of public transport usage, densities of travelers in stations and vehicles, flow, etc., information that is very useful for public transport managers. It must be stated from the beginning that the goal of this research is not the precise location of travelers, but instead a solution to acquire information regarding traveler densities and flow. The approach is intended to offer a low-cost, low-energy and eco-friendly solution. Increasing the interest of people in public transport will decrease private vehicle traffic, thus reducing travel stress, environmental pollution, and noise in large urban agglomerations. Key findings of the research are as follows:Field measurements proved that representativity of discoverable devices in relation to the real number of persons allows for a convenient estimation of real-world traffic behavior.Experiments on received signal strength and other transmission parameters for different scenarios have emphasized the main influencing factors that affect precision.Proposal of adequate models and solutions to improve the accuracy of location and tracking.

Recommendations for decreasing the energy consumption of the sensing infrastructure for public transport are the result. There is a growing interest in determining new, less costly, and more effective technologies to enrich the information collected in support of public transport management.

Liu, H.; Darabi, H.; Banerjee, P. and Liu, J. present in [1] an overview of the currently developed wireless indoor positioning solutions and classify different techniques and systems. Compared to similar work, new scenarios and conditions are evaluated in the present research, e.g., the influence of hardware diversity on discoverability, and the ages of people on representativity, when employing BT detection for traveler flow evaluation.

Farid Z. et al. [2] propose a review of the recent advances in wireless indoor localization techniques.

In [3], the authors propose a Voronoi and crowdsourcing-based mapping interpolation for general regression neural network fingerprinting localization, employing WLAN. The solution appears to be simple and efficient in terms of energy consumption and hardware deployment but has the disadvantages of only using Wi-Fi signal propagation, and the necessity of employing specific access points.

Liu J. et al. [4] investigate smartphone positioning as an enabling technology in the navigation and mobile location-based services (LBS) industries. The solution employs smartphone sensors to determine the motion dynamics of the user, two algorithms based on hidden Markov model (HMM), a grid-based filter and a Viterbi algorithm for data fusion for improving the position estimates of the users.

Two novel BLE-based localization schemes, namely Low-precision Indoor Localization (LIL) and High-precision Indoor Localization (HIL) are described in [5], based on RSSI information collection, to create a specific region in space, where the target is located with high probability.

Trilateration and fingerprinting based on Wi-Fi are technologies that the authors of [6] use as an affordable means for indoor localization. The researchers employ a combination of two methods, involving the use of matched pre-recorded received signal strength (RSS) from known-location access points (AP) to the data transmitted from the user on the fly, and a distance-based trilateration approach using three known AP coordinates detected on the user’s device to derive the position.

The study [7] proposes 3D Voronoi diagrams for analyzing and visualizing 3D points instead of the original data item.

[8] Mahtab Hossain and Wee-Seng Soh [8] present a comprehensive study on Bluetooth radio signals parameters such as RSSI, Link Quality, Received and Transmit Power Level to determine its usability for location services. They demonstrate that RSSI correlates satisfactorily with distance and recommend it as “the most desirable Bluetooth signal parameter to be used in location systems”. On the other hand, Bahl P. and Padmanabhan V. describe a system that uses a combination of empirical measurements with signal propagation modeling to determine user location, based on the storage and processing of signal strength information from multiple base stations [9].

In [10], the authors combine several RF technologies in two location algorithms, namely SELFLOC, and RoC, which tries to solve the problem of aliasing the signal. The communication technologies employed in this research are WLAN and Bluetooth. The accuracy obtained for stationary users reached 1.6 m.

In the medical domain, research presented in [11] employs location services for mobile biomedical and/or industrial sensors that are Bluetooth enabled, based on RSSI measurement for distance estimation, combined with a triangulation mechanism similar to GNSS.

There are many papers and research studies that propose RSSI or RX as a primary information source, which leads to the conclusion that this represents a feasible and economic instrument in indoor location techniques [12]. On the other hand, solely relying on the RSSI metric has proven to be less precise in localization techniques than a combination of different technologies [13]. The authors of [14] consider that, although a broad spectrum of algorithms can trade accuracy for precision, none brings a significant advantage in localization performance. In [15] the authors perform an analysis regarding different aspects of ubiquitous LBS technologies, multi-radio localization and seamless positioning, and a similar approach is presented in [16]. An analysis on the multipath propagation of radio signals for indoor positioning services is given in [17], where a solution of “beams (formation) by sequentially switching through each element of an antenna array and forming the beam with phase and gain corrections in the receiver’s individual channel correlators, using only one RF front-end” is employed. Similar analysis is given in [18]. The cellular network is used for indoor location applications with lower precision, as presented in [19].

A connection with transport applications is given in [20], where Zhao Y. considers the solution of cellular-based location services as having a potential impact on future intelligent transportation systems, including telematics and public transit systems.

There are also other media for obtaining location information, such as ultrasonic waves. Moreover, this approach offers a privacy feature, provided the users are able to use ultrasonic devices and that they are able to find their own location by themselves [21]. In [22], the authors employ FM broadcast radio signals for robust indoor fingerprinting.

Another solution tested by some researchers uses active RFID tags [23] in a controlled area surrounded by nine readers.

Similar research has been pointed out in [24], where the authors perform a spatial localization system for networked devices based on ambient sound sensing. The system is based on TDoA (Time Difference of Arrival), and is able to locate a position in a sound-controlled area.

The solution of [25] claims that it is capable of tracking the movement of a smartphone relative to the acoustic direction with a resolution of less than 1 mm when the device is shaken. “For indoor localization, the 90-percentile errors are under 0.92 m. For real-time tracking, the errors are within 0.4 m for walks of 51 m.” This represents an interesting solution for indoor localization, but it will probably need a map of the inner environment, and also to employ techniques for navigating such as “dead reckoning”, which have the disadvantage of cumulatively adding errors in longer travels.

Lasers for indoor localization and mapping in a scoping review of different positioning technologies for indoor environments are presented in [26].

Finally, several papers present in the domain’s literature use data fusion as a best solution for increasing accuracy in the localization and indoor tracking of different objects or persons [27,28,29].

To conclude, the subject of indoors localization appears to be of interest to and benefiting from many research activities. For public transport management and infrastructure management, Bluetooth appears as a very attractive technology due to the following advantages:In present developed urban environments exist enough devices with enabled Bluetooth, owned by travelers and capable of being detected and/or tracked,BT has a low consumption of energy,Precise enough indoor localization based on RSSI, signal strength and/or other parameters is also suitable for the movement tracking of people, passenger flow evaluation, or origin–destination estimation.

The remainder of the article is organized as follows: Section 2 describes Materials and Methods—usage of hardware and software specific tools to determine in the field the representativity of discoverable Bluetooth and Wi-Fi devices carried by people in selected subway stations, in comparison to the real number of physical persons in the regions of interest. Section 3 presents the results of various tests involving different scenarios that may be encountered in the real world in relation to the data collection methodology presented in this research. Section 4 (Discussion) emphasizes the main findings from the experiments and data processing, and finally Section 5 (Conclusions) presents the overall image of the solutions and future development.

## 2. Materials and Methods

GPS is the technology usually employed when trying to determine a target, such as a traveler/passenger’s location in outdoor public transportation. Things change significantly when indoors, where GPS signals are extremely weak or completely missing. Still, there remains the possibility of using GSM triangulation, where available, with lower precision. The detection of passenger flows and densities of people is useful in satisfying transport demand and detecting abnormal situations. In this paper, research is conducted to determine the usability of Bluetooth and Wi-Fi technologies for that purpose. The goal is to see if enough information can be gathered, considering a post-processing phase to determine, with sufficient approximation, real-world behavior. Usually, this solution does not require dedicated detection equipment in the field, but only a simple placement of Wi-Fi access points, Bluetooth detectors on the paths of travelers, or on public transport vehicles. The methodology consists in counting MAC addresses, along with timestamps at the location of the detector. For more precise positioning, it is possible to consider specific configurations for the BT detectors, able to perform post-detection operations such as triangulation or trilateration. This solution is more suitable for the placement of sensors in the public transport vehicles, as suggested in previous works [30,31]. Bluetooth, as an emerging technology in this direction, compared to other communication technologies, has the advantage of strong popularity and deployment throughout the mass of travelers. Table 1 shows in a comparative mode the main characteristics of Bluetooth, in comparison to similar technologies.

The present paper extends this research, trying to establish the efficacity of anonymous detection of traveler flows in various conditions, including heterogeneity of hardware, influence of the presence of people, positions of mobile devices, attenuation of different obstacles, and finally recommending solutions and models to improve precision.

### 2.1. Field Tests for Determining the Representativity of Detectable Devices among People and/or Vehicles

A first step in the process of evaluating the efficiency of the proposed methodology is to determine the ratio between the detectable devices carried by, and the physical number of persons in, the region of interest (ROI). As demonstrated in previous works [32,33], the behavior of Wi-Fi and Bluetooth signals differ essentially in subway tunnels and indoor platforms from those from open-field conditions. Moreover, the presence of people in the direct path of propagation may also influence the internal chip’s received signal strength indicator, or signal strength if these metrics are used as an indicator of the distance between the sensor and the target. It has been considered useful to present in this analysis some findings of the behavior of received signals in different operating situations that may be encountered in the real-world.

#### 2.1.1. Setups of Subway Testing Environment

One of the most frequented places in public transport is the subway. For the purpose of establishing the representativity of detectable BT devices and the real number of travelers, several experiments have been undertaken in several subway stations and trains.

The local subway network is 750 V DC powered, fed through a third rail. Other equipment that may induce electromagnetic influences in communications include track circuits, ATC (Automatic Train Control), or CBTC (Communications-Based Train Control). Stations and platforms are fitted with GSM microcells for all four local mobile communications operators, and even some Wi-Fi access points for some of them. With very few exceptions, all stations include pillars. These cause additional reflection and attenuation of radio signals. In these cases, two or more ray signal propagation models may be necessary for the exact determination of locations and distances to received targets. Figure 1 and Figure 2 show the environment setups where the first tests have been performed.

Materials used for testing:Hardware: Mobile phones Realme RMX3511 (C35)—Android 11, Xiaomi Redmi Note 9Pro—Android 11/MIUI 12.5.8, Samsung SM-A505FN/DS (A50)—Android 11/UI 3.1,Software: Bluepixel Technologies LLP/BLE Scanner version 3.21, BLE Analyzer version 1.1 /June 10,2020, BLE Radar V1.0.Methods for counting people: direct observation and recording.

#### 2.1.2. Purpose of Tests

The goal of the field tests in this phase has been to determine the ratio between the number of detectable BT and/or Wi-Fi devices and the real number of persons in the ROI. Tests have been conducted in different conditions:On the platforms of the subway, with no trains in station.On the platforms of the subway either with one or two trains in station.In trains, in movement, in tunnels.

It is worth mentioning that only detectable devices may be considered as targets in these tests. It has been found that the number of connected devices (wearables) are even larger than the discoverable smartphones, and usually a person that carries a connected smartwatch or BT phone is more likely to be detected than a carrier of a simple smartphone.

#### 2.1.3. Results of Field Measurements


Conditions of first test: weekend day—station not busy.


University Subway station (named here Subway Station 1) was chosen as the first site for making field evaluations, as it is one of the busiest stations. Table 2 presents figures determined.

In Figure 3 the number of detected devices is represented accordingly versus the real number of persons in the station. In most cases, with few exceptions, the detected number of devices follows the general trend of persons present on the platforms.

Another series of similar tests have been conducted in the same conditions and with the same equipment in another busy subway station (Obor), named Subway Station 2. The results are given in Table 3.

Figure 4 presents the evolution of detected devices in comparison with the real number of persons in the station for the second experiment.

For Subway Station 2, the average representativity of detectable devices amongst physical persons reached the value of 31.604%. One observation that is worth mentioning is the influence of the age of travelers. It has been noticed in field experiments that when the proportion of elderly people is dominant versus younger persons, the detectable number of devices is less representative of the real number of persons waiting on platforms.

The above results were obtained in conditions where no train was in the station. The same measurements were also taken in the presence of trains. These results are presented in the diagrams below—Figure 5 and Figure 6.

The representativity percentages recorded were, accordingly, 36.77% for Subway Station 1 and 30.58% for Subway Station 2. With two trains simultaneously arriving at the station, the detection percentage reached 36.41%. These values generally oscillate between 14 and 20 percent, depending on the categories of travelers’ ages and affinity towards the use of mobile devices. Note that the exact number of passengers when one or two trains arrive at the station simultaneously is only estimated.

The second series of field experiments were conducted to determine the number of detectable devices in trains when traveling in tunnels, so no external influence of APs or other Wi-Fi devices is located nearby. It has been observed that the signals experience better propagation in tunnels; therefore, this characteristic behavior should be considered when estimating the number of onboard travelers, based on the discovered BT devices. During the hours where tests were performed in the field, the average number of travelers per carriage was 20. It is also important to mention that BT signals can also propagate from neighboring carriages, so the real number of physically present persons should also take these ones into consideration. Figure 7 shows the recorded results for this specific case.

The average ratio of detectable devices over the real number of travelers in trains reached 48.91%. Since most people use their smartphones intensively during travel on the subway, there is a higher detection percentage.

### 2.2. Wireless and Unconventional Sensing Technologies—A Basis for Smart Mobility

Smart mobility is a leading strategy in developing modern urban agglomerations.

The present research proposes a solution to employ “moving sensors”, with the travelers and traffic participants themselves becoming sensors in traffic. This section is dedicated to an assessment of different conditions in which Bluetooth and/or Wi-Fi may become possible candidates for a wide range of sensing technologies with the potential to be used for gathering information regarding traffic and travel flowing in large networks, both outdoors and indoors.

#### RX—Based Location of Travelers

RSSI, or Received Signal Strength Indicator, is a measure of a signal’s strength at a certain point of a transmitter (an integer value in a register), being used as a metric for several types of technologies for short- and medium-distance communications. RX is the absolute value relative to 1 mW. The RSSI can be converted into dBm using the different relationships (for example, in the 433 MHz band used for powering the IoT sensor networks, where a node, or mote, provides RSSI on the Analog to Digital Converter (ADC) channel 0 and is available to the software running on the mote as a 10-bit number [34]:(1)$$V_{RSSI}=V_{batt}\cdot ADC_{COUNTS}/24$$
(2)$$RSSI\;\left[\mathrm{dBm}\right]=-51.3\cdot V_{RSSI}-49.2$$
(3)$$SS\;\left[\mathrm{dBm}\right]=10log_{10}\frac{P\left[\mathrm{mW}\right]}{1\mathrm{mW}}$$
where SS means signal strength, P—power, and $V_{RSSI}$ is ranging between 0 and 1.2 V, with higher voltage meaning lower input signal.

Signal propagation issues such as reflection, refraction, and multi-path cause the signal attenuation to correlate weakly, leading to less precise distance estimations. Because signal strength decreases in a non-linear and mostly unpredictable manner with distance, a map of the recorded signal strength values is usually needed and pre-defined locations must be created first. This phase is generally labor intensive and reduces the applicability only to pre-mapped locations. To reduce uncertainty regarding the exact position determined with this technique, further statistical filtering and error minimization algorithms must be applied.

Signal strengths may be mapped along the distance between the transmitter and the receiver according to the combined effects of large-scale fading (travel of the signal on a long distance and effects of absorption by the objects on the path) and small-scale fading (travel of the signal via multiple paths and their effect at the reception site) [35]. Usually, the large-scale propagation section is modelled by the means of RSS and employs a log-normal distribution, while the small-scale propagation component employs Rayleigh distribution (for NLOS cases), and Rician distribution (LOS cases). Commonly used filters for the attenuation of RSS fluctuations include moving averaging [36], exponential averaging [37], and Gaussian filtering [38].

The Wi-Fi signal strength measurement is not a very accurate metric for indoor localization. It can reach between 30% to 60% variations for the same location and conditions of propagation; therefore, several measurements and averaging or finding a centroid are needed in the post-processing of information to obtain better results. Signal strength may vary between −100 and 0 dBm. Indoor environments always create difficult propagation conditions, strongly different from the direct line of sight propagation in a free environment.

BT/BLE technologies also experience variations of signal strength, due to external factors influencing radio waves—such as absorption, interference, or diffraction, making this type of indoor location technically difficult without adequate filtering of results and several measurements performed to improve the accuracy [31].

A typical formula for determining the approximate distance to the transmitter (FOV) is given by:(4)$$d=10^{\left(P_{m}-RSSI\right)/10N}$$
where d represents the distance [m], $P_{m}$—the measured power (or the 1 Meter RSSI indicator), and N is an integer constant, depending on the strength of the signal. The usual value for N is 2.

In this section, several aspects of BT and hardware conditions will be investigated to determine its usability for tracking people and vehicles and determining the density of targets and/or traffic flow.

## 3. Results

A series of tests have been conducted in specific conditions to determine the usability of signal strength and RTT (round trip time) solutions in tracking the mobility of people in transport terminals. The tests had the purpose of establishing to what extant different operation conditions, positions, environment, and hardware diversity influence accuracy in position and movement tracking evaluation. This section describes in more detail some of the conducted experiments.

### 3.1. Setups for Experimental Analysis on Signal Strength Usability for Travelers’ Tracking Purposes

A specific structured indoor environment was set up in a laboratory to determine the accuracy of received signal strength for distance measurement (Figure 8). The communications technology employed was Wi-Fi IEEE 802.11n. A Huawei fixed AP and a Xiaomi Note 9T were employed as different APs from where the mobile device (Xiaomi Note 9Pro) sent ping messages, recording values of signal strengths and data speeds.

The Wi-Fi channel used was 1, frequency 2412 MHz.

### 3.2. Results of Experiments

Considering the conditions of testing described above, the results obtained are presented in Table 4:

Filtering has been applied to the data, retaining only values between a previously established threshold error distribution (from an average below 70%). The results summarized in the above table are given in Figure 7, a diagram showing variations of the signal strength over distance, and absolute error over distance is given in Figure 9. It can be noticed that the signal strength level experiences significant variations, especially when in NLoS (Non-line-of-sight) conditions, a fact that is likely to be encountered in many situations in indoor environments.

Signal strength levels decrease with distance, quasi-linearly after the first meter, up to around five meters, then decrease abruptly when direct visibility to the AP disappears. After a certain point, where NloS occurs, the variation increases and precision decreases significantly more (Figure 8).

To determine if there is also a dependency on the hardware, the experiment was repeated in the same environmental conditions, employing as the AP a Xiaomi Note9T smartphone, Wi-Fi channel 13, frequency 2472 MHz (Table 5).

In this case, there is more linearity in signal strength variation with the distance than in the previous case, which leads to the conclusion that another factor that must be accounted for in these techniques of indoor localization is the hardware (and associated antennae) diversity—Figure 10.

From this sequence of tests, the following conclusions may be drawn:The variations of signals strength indications after certain distances to access points are increasing.Hardware diversity may also induce variations in distance measurements based on Wi-Fi technology.

To alleviate the efficiency of this location methodology, it is recommended to ensure averaging of several RSSI determinations for mobiles with low dynamics and to introduce supplementary correction factors, which may be determined on site by experimental actions.

### 3.3. Experimental Analysis on Wi-Fi Signal Strength Variation in Stationary Conditions—LoS Connection

To assess the stability of Wi-Fi signals over time, at a fixed location and in stationary conditions, a test has been performed for 100 measurements (Figure 11). The distance from the AP was established at 2.1 m.

Equipment employed:AP: Huawei Technologies, MAC 58:20:59:71:B4:FD, WPA2,Wi-Fi channel 1, f = 2412 MHz, channel width 20 MHz, Link speed 65 MbpsMS Device: Xiaomi Redmi Note 8 Pro, MIUI Global 12.0.5, Android 10 QP1ASoftware: Measurement: Network Signal Info version 5.74.03, Ping Tools 4.64 Free, Wi-Fi Analyzer V1.0.4, Signal Strength V26.1.1. Data processing: Excel version 2211 (Build 15831.20122), Weka 3.8.6 (Weka Environment for Knowledge Analysis).

Data have been collected for 100 signal strength readings, LoS, indoors environment. The distance from the AP has been computed employing the following formula:(5)$$D_{AP}=10^{\frac{P_{M}-RSSI}{10\cdot N}}$$
where $D_{AP}$ represents the computed distance to the AP according to the internally determined RSSI, $P_{M}$ is a power measurement level, and *N* represents a constant depending on the environment (in this case, the value considered for N is 3). Dispersion of resulted values is presented in Figure 12. For data processing, Weka 3.8.6 was employed to determine the different parameters.

The lower part of Figure 12 shows the acquired distance distribution— 44% of the results were correlated with the real distance to the AP, the rest extending towards an error of approximately 0.5 m in location accuracy. The conditions of the test included direct line of sight (LoS) to the AP and usual interferences in an indoor environment (few access points and BT devices active). As can be noticed in the above diagram, the error in distance comprises between a few centimeters and about 0.8 m, which means that the location process needs to be corrected via different averaging and weighting methodologies.

The same test has been repeated with the same hardware, but different software for computing the distance to the access point (for Wi-Fi, channel 1).

In Figure 13, most of the values are relatively close to the real distance—however, there still are variations (periodical, as observed experimentally, probably an effect similar to fading). The collected set of data has been classified according to the EM classification algorithm (in Weka software). The results are presented in Figure 14. Four clusters resulted comprising the most relevant group of data.

During the tests, a certain degree of repeatability was noticed according to a pattern that may be suitable for modeling, to predict deterministic variations of the signal strength and to further improve the detection and ranging accuracy.

Conclusions regarding this sequence of tests: there is variability in the received signal strength information in stationary conditions, which may be due both to the hardware and propagation conditions, including here the density of the neighboring AP and BT devices. Averaging measurements may improve accuracy.

### 3.4. Experimental Analysis on Wi-Fi and BT Signal Strength Variation in Stationary Conditions—NLoS Connection

To determine the influence of an armored concrete wall on distance measurement precision employing Wi-Fi signal strength as a measurement parameter, an experiment was conducted, maintaining the same conditions as in the previous sequence of tests, with the exception of changing the position of the receiver behind a concrete-armored wall of 20 cm thickness (Figure 15).

The same hardware was employed; software: Speed Test Master Pro Ver. 1.44.0. Data processing: collection of results in Excel file, conversion to ***.csv format, conversion to ***.arff file, running Weka software with Linear Regression classifier and clustering with Expectation Maximization algorithm. The results are presented below (Figure 16).

Instances: 98. Correlation coefficient: 0.9745, Mean absolute error: 0.075.

In this case, it is noted that the error in the predicted distance is significantly larger, varying from 1.81 m to 5.75 m (Figure 16), which leads to the conclusion that for Wi-Fi RSSI-based distance and location processing, supplementary actions are needed to improve accuracy (such as prior signal fingerprinting of the area).

The precision in signal strength-based distance measurement is significantly decreasing in the case of NLoS, as the clustered groups of data show, where most measurements indicate a distance to AP around 50% larger than the real one when the radio signals are transiting an armored concrete wall of 20 cm thickness.

Comparison with Bluetooth technology—similar measurements have been performed for a Bluetooth device, in the same conditions. According to the results, the BT technology is influenced in a significant way by variations in signal strength. Equipment employed:BT devices: Amazfit Bip Watch,Xiaomi Redmi Note 8 Pro, MIUI Global 12.0.5, Android 10 QP1ASoftware: Measurement: BLE Analyzer. Data processing: Excel, Weka 3.8.6 (Weka Environment for Knowledge Analysis). Weka Environment analysis (with Linear Regression classifier)—Figure 17:
Correlation coefficient 0.9961Mean absolute error 0.0255Root-mean-squared error 0.0291Relative absolute error 9.6277%Root relative squared error 8.8569%Total Number of Instances 98.

As can be noticed in Figure 17, the collected data are more dispersed around the correct value in the case of BT RSSI measurement, compared with the Wi-Fi, where data are more clustered, which gives more uncertainty and imprecision in the BT detection. However, since BT’s range is shorter than Wi-Fi’s, the uncertainty in detecting distant mobile stations is reduced.

### 3.5. Influence of Travelers’ Density in Signals Propagation

In large agglomerations, such as in metro stations or other transport terminals, another factor that is to be taken into consideration in the detection of traveler flow is the mass of people, which may also have influence on the location precision due to the signals being absorbed by human bodies. In previous research [30], it has been demonstrated that “the RSSI readings are rapidly dropping in the vicinity of the transmitter, then the variation is slowing down up to distances larger than 8 m and it appears the field keeps in a relative slow variation around −80 dBm.” Furthermore, the density of people may lead to additional absorption of radio signals. However, accuracy can be significantly improved if preliminary radio mapping of the region of interest is performed and specific algorithms for matching received signals’ strengths are used, such as KNN (K-nearest neighbor). In case a subscription of the users is available to a location and route guidance service, along with the signal strength location method, Round Travel Time (RTT) may also be used to enhance location precision. The following section discusses these alternatives.

### 3.6. Enhancing Indoors Location Accuracy with Additional RTT Computing

A solution to improve accuracy would be to add a supplementary localization process via measurements of round-trip time (RTT) [34]. For this to be achieved, it is necessary that the mobile station is enrolled in an application that is synchronized with the infrastructure of APs or BT stations, and to exchange periodic beacon messages to a pair, at least, of fixed anchor points. Then, a process of triangulation/trilateration is used in combination with signal strength measurement to determine the location of and range to the detected device. This technology has some advantages from both the technical and economic points of view: it has ubiquitous coverage due to many access points and ease of installation, needs no further hardware, the range is sufficient for indoor applications (approx. 50–100 m), and it does not have so many restrictions regarding line of sight comparatively with other, similar location techniques.

In time-based distance estimation, a data packet (ping) is transmitted between the AP and the targeted device and RTT is determined afterwards, using the following relationship:(6)$$d=\frac{v_{r}\cdot RTT}{2}$$
where $v_{r}$ represents the speed of radio waves in air, in the conditions of the measurement location environment. Due to minimal difference between the actual radio waves’ speed and light, in general applications we may consider that $v_{r}≅c$. The most annoying problem in this technique is, however, the synchronization between the two connected devices, which makes Wi-Fi applications for indoor localization prone to errors. Therefore, it is proposed to employ one of these two methods:

For precise location applications with high accuracy, it is recommended to make use of GPS-received UTC synchronization and time stamping of messages (limited application for mass development in indoor locations), and for statistical data collection or movement tracking indoors, use of multiple measurements, followed by statistical filtering and clustering algorithms for reducing inaccuracy.

However, for better location precision, it is necessary to also consider the time needed by the receiving device to process the information, $\tau_{p}$, and to send message back to the interrogator:(7)$$d=\frac{v_{r}\cdot \left(RTT-\tau_{p}\right)}{2}$$

Because $v_{r}\cdot \tau_{p}=d_{p}$, (where $d_{p}$ represents a distance equivalent to the time needed for processing the message), the result is that there is a minimum range of $d_{p}$ below which it is not possible to measure the distance to the target device with the desired precision. Therefore, for a more accurate determination, in this case, it is necessary to estimate these elements firstly by placing the two connected devices nearby (distance = 0) to establish the zero-distance reference (which corresponds mainly to the processing time-equivalent distance $d_{p}$). The advantage of RTT is that it is a process that does not require synchronization, and it can be easily implemented in mobile applications.

Experimental results: in an indoor environment, experiments with two connected devices have been carried out, with one being set as an AP and the other as connected mobile station. The mobile station sent a series of ping messages to the AP, recording time values. The results are presented in the following table, with the first measurement at a zero-meter distance. The duration of the test was 60 s, with 58–59 packets sent and a reception packet loss varying from 0% to 1%. The recorded values presented below (Table 6) were taken during approximately 60 s for every meter of distance between the AP and the mobile device.

According to the obtained results, it can be noticed that in the case of non-synchronous procedures employing ping in usual Wi-Fi networks, the round-trip time method is still quite imprecise (Figure 18), having large variations of RTT in time due to propagation effects and the presence of other Wi-Fi devices in the area.

LoS and NLoS cases have significant influence on the RTT, so these special cases should be taken into consideration. However, by using path analysis with statistical filtering and clustering algorithms, precision can be improved. When trying to assess the propagation characteristics in indoor environments, the following models are more suitable:TWGRM—Two-Way Ground-Reflection Model, which is applicable to situations where the two (transmitting and receiving) antennas are in line of sight (LoS). The model assumes that the detected device antenna receives both a direct line of propagation signal and a ground-reflected (delayed) signal. This model may be used for computing the expected distance in specific outdoors environments.LDPLM—Log Distance Path Loss Model, which is the most suitable model for densely populated areas and industrial environments (as is the real situation in crowded cities)

(8)$$PL_{LDPLM}\left(d\right)=10lg\frac{P_{Tx}}{P_{1}}-10lg\frac{P_{Rx}}{P_{1}}=PL_{0}+10\gamma lg\frac{d}{d_{0}}X_{G}$$
where:

$PL_{LDPLM}\left(d\right)$—the path loss model [dB]

$P_{Tx}$—the transmitting antenna power

$P_{Rx}$—the power received by the recipient device
$$P_{1}=1\;\mathrm{mW}$$

$PL_{0}$—the path loss at the reference distance $d_{0}$, based on the Friis model for free-space path loss

γ—the path loss exponent

*d*—length of the signal path

$d_{0}$—the reference distance (10 m for BT)

$X_{G}$—expression of attenuation via a random normal/Gaussian variable [dB].


WINNER II Indoor Model—for indoor scenarios:


(9)$$PL_{W-II}\left(d\right)=43.8+36.8lg\left(d\right)+20lg\left(\frac{f}{5000}\right)+X+\left[17+4\left(n_{f}-1\right)\right]$$
where:

*d*—path length [m], $d\in \left(3,\ldots ,100\right)$

*f*—frequency, $f\in \left(2,\ldots ,\;6\right)$ GHz

$n_{f}$—the number of floors in the building

*X*—expression of attenuation due to wall pass through of signal

$X=5\left(n_{w}-1\right)$ for thin walls

$X=12\left(n_{w}-1\right)$ for thick walls.

### 3.7. Analyzing the Influence of Hardware Diversity in Indoors Localization Based on RSSI//RTT Technologies

Another problem that must be taken into consideration when proposing a service for indoor localization based on the RSSI/RTT technologies described above is the diversity of hardware equipment owned by tracked travelers. This diversity might manifest itself in variation in the nature of devices (smartphones, smartwatches, Bluetooth headphones), equipment sensitivity, transmitting power, processing times, or other similar characteristics. To determine to what amount this problem may affect the efficiency of the proposed solution, a series of tests have been conducted using different types of smartphones. The results, however interesting, are presented below.

The purpose of the test is to determine the influence of mobile station hardware on received signal strength. The BT transmitter has been placed at the distance of 1 m from the receiver, direct field of view, and received signal levels were recorded for 100 times from four different smartphones: Huawei Mate 20 Lite, Samsung Galaxy A50, Motorola Moto Z3, and Samsung Galaxy A12. The results, however interesting, are presented in Figure 19, Figure 20, Figure 21 and Figure 22.

Conclusion regarding the results of the test sequence: the hardware has a very important contribution to obtaining accurate measurements. From the tests performed, it appears that there are quite significant variations in BT modules’ sensitivities to signal variations, and the experiments showed that Moto Z3 had the best stability in presenting the correct signal levels (over 60% of the samplings were correctly approximating the real RSSI—Figure 21, Histogram 3); at the opposite end was the Huawei Mate 20 lite, with only 23% values closer to the real one (Figure 19, Histogram 1) and almost all values lower than normal, which denotes an overall reduced sensitivity of the BT-embedded module. The other two mobile terminals showed intermediate results.

For improving the accuracy of those devices that showed good performance, a Markov Chains model is proposed, which may be used to predict states of the hardware for applying corrections to the received signal strength indicator obtained. The Markov Chains model has been constructed for the Motorola smartphone, based on the collected data above. It appears that most of the transitions occur around −55…−56 dBm, with this equipment showing excellent stability over time in signal reception, compared to other devices. Based on the obtained results, it may be concluded that different equipment induces supplementary variation around the correct signal strength value (with range differing from device to device). This ascertainment imposes a very careful approach for the post-data-collection filtering techniques, to regularize the results. A regularized histogram for Motorola Z3 sensitivity in a scenario under stationary conditions is presented in the following diagram Figure 23.

The associated Markov model is presented in Figure 24.

The final series of experiments was conducted by placing the hardware at the distance of 2 m from the transmitter in direct view, and signal strength levels were recorded 100 times. Obtained data have been converted into a CSV file, pre-processed, and then analyzed with Weka software to determine the dispersion of recorded values (Figure 25).

The BT transmitter has been placed behind an armored concrete wall (metal bars), and the RSSI levels were recorded 100 times (same setup as in Figure 15). Obtained data have been converted into a CSV file, pre-processed, and then analyzed with Weka software (Figure 26). It can be observed that the concrete wall induces more unevenness in the recorded values of RSSI, a fact that involves the necessity of averaging several measurements to obtain a more reliable RSSI value.

As an overall conclusion regarding this last sequence of experiments, a better target coarse position acquisition may be determined by performing initial measurements and model construction based on Markov Chains in the case of fixed BT anchor points. This method is proposed for initial geofencing of the coverage area and initialization of the indoor localization process. It can be concluded that efficient averaging and filtering processes for the recorded RSS levels are highly recommendable to improve precision for indoor localization scenarios where the environment, along with the diversity of equipment, might induce significant variance in RSS.

## 4. Discussion

The present research was aimed to determining relevant aspects of traveler flow information collection employing detection of discoverable Bluetooth and Wi-Fi devices. The following aspects were in focus:Representativity of discoverable devices in the mass of physical persons present in the region of interest.Analysis of different scenarios that may influence the accuracy of the collected data: behavior of received signal strength under LoS and NLoS conditions of propagation, influence of concrete walls on signal strength, influence of positions of terminals and density of people on signal strength (presented in [30]), influence of travelers’ ages on representativity, and influence of hardware diversity.Possibility to determine data correction solutions and behavior models for specific hardware to improve the accuracy of data.

Based on the conducted field experiments presented in this work, it can be mentioned that process of collecting information via indirect sensing, i.e., employing detection of active, discoverable devices using Bluetooth, or Wi-Fi, is not recommendable for precise measurements or counting of people and/or vehicles. However, the goal of this research was to determine if this methodology is suitable for obtaining relevant information regarding densities of passengers, fluency of traffic, or in the prediction of travel demand. For these purposes, it may be concluded that the technology is indeed suitable and can offer a very cheap, environmentally friendly, and efficient method for obtaining trends and overall images of the traffic and travel patterns of interest.

Other aspects that are worth mentioning here are the security and privacy of collected data. Because MAC addresses of BT and / or Wi-Fi devices are detectable and recordable, specific data protection measures must be taken to make it impossible to connect a specific MAC address with the person that carries the respective device in any way. This can be accomplished via recording of temporary tokens, or specific primary keys, instead of the exact MAC discovered address, then removing these when data present no more interest. The latest developments in Android-enabled devices include MAC address randomization capability, which makes tracking difficult, but does not affect measurement of traveler densities for transport demand analysis [39,40,41].

The clustering of data is also very important in the post-process analysis of imprecise information. The main goal of this phase is to extract features that are relevant in traffic and transport flow analysis.

Clustering and classification of collected information result in secondary, relevant information on the development of traffic flow, pedestrians’ affluence and densities, flow generation and flows absorption locations in public transportation. Notable features of collected data may include:Variability in time and space.Variability in representativity in comparison to the whole set of individuals, due to the data collection methodology employed.Non-uniformity in the density and shape of collected data points.Variability of interest parameters associated with the collected data.

## 5. Conclusions

The activity of collecting information regarding passenger and traveler flow in large transport terminals, metro stations and public transport vehicles is not an easy task to put in practice without large amounts of materials and energy and the deployment of many sensors in the network. This paper has investigated the usability of simpler technology for data collection, including anonymous detection of active, discoverable Bluetooth and Wi-Fi devices carried by persons in these locations. Field measurements in different conditions have been conducted to determine the usability of this solution and the ratio between the discoverable devices and the real number of physical persons in the regions of interest. Based on these results, and on the forecasted trends in the evolution of smart devices, IoT and 5 G, it can be concluded that the number of discoverable elements offers sufficient mass nowadays to develop this solution on a large scale, accompanied by efficient post-processing of big data collection and specific clustering algorithms. As a future development, it is intended to expand the research to complex detection of both travelers and vehicles and post-processing analysis by employing BT sensors mounted on public transport vehicles and subway trains.

## 6. Patents

The research, experiments and data processing partially presented in this article represent a continuation of the research for the patent: M. Minea, C.M. Dumitrescu, I.C. Chiva, V.L. Minea, A. Semenescu. “Method for anonymously acquiring position and mobility data in public passenger transportation unit, involving categorizing received nodes and unit for statistical analysis of four categories of Bluetooth/Bluetooth Low Energy devices”. RO134415-B1/RO134415-A0, DIIDW:2020857359, 2022.

## Figures and Tables

**Figure 1 sensors-22-09025-f001:**
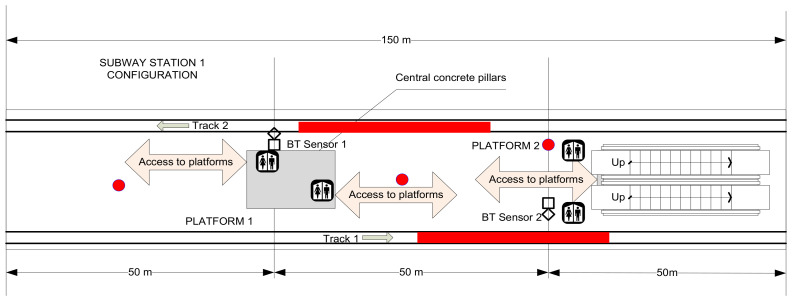
Aspect of Station 1 test bed setup (red dots representing placement of human observers).

**Figure 2 sensors-22-09025-f002:**
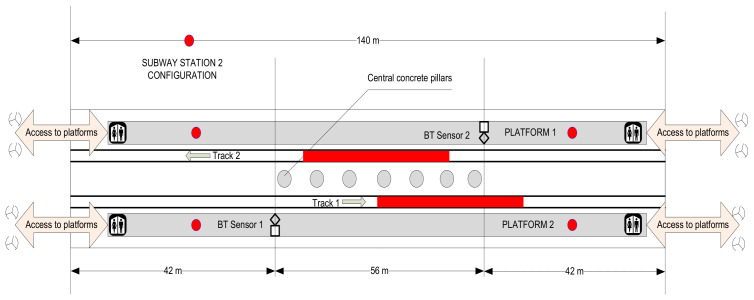
Aspect of Station 2 test bed setup (red dots represent placement of human observers).

**Figure 3 sensors-22-09025-f003:**
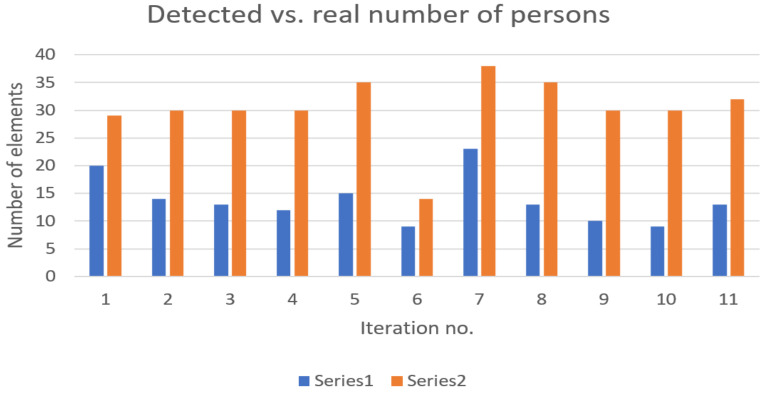
Detected devices (represented in blue—Series 1) versus real number of persons (represented in orange—Series 2) present on platforms—no trains in station (Subway Station 1).

**Figure 4 sensors-22-09025-f004:**
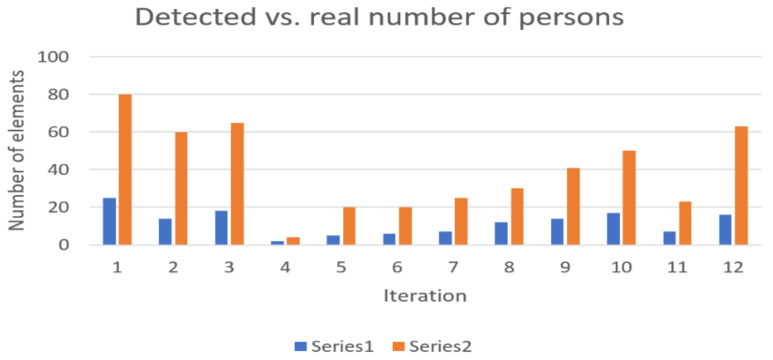
Detected devices (represented in blue—Series 1) versus real number of persons (represented in orange—Series 2) present on platforms—no trains in station (Subway Station 2 subway station).

**Figure 5 sensors-22-09025-f005:**
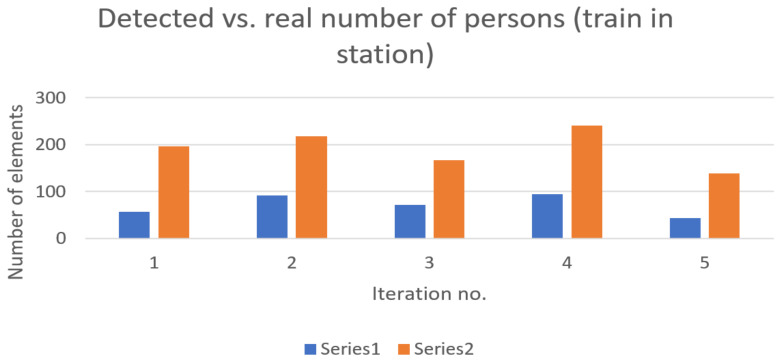
Detected devices (represented in blue—Series 1) versus real number of persons (represented in orange—Series 2) present on platforms—one train in station (Subway Station 1).

**Figure 6 sensors-22-09025-f006:**
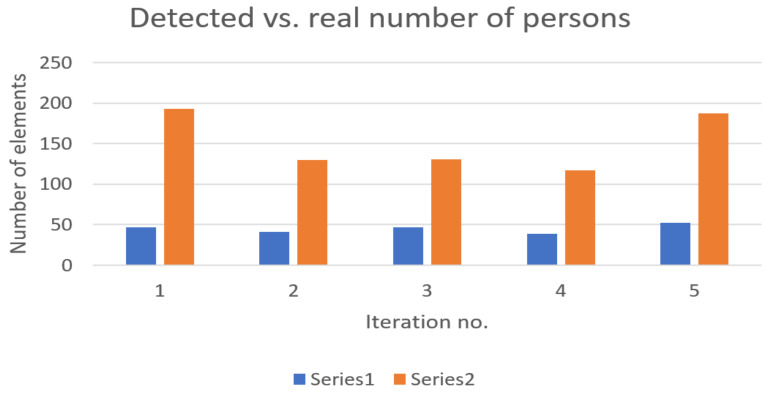
Detected devices (represented in blue—Series 1) versus real number of persons (represented in orange—Series 2) present on platforms—one train in station (Subway Station 2).

**Figure 7 sensors-22-09025-f007:**
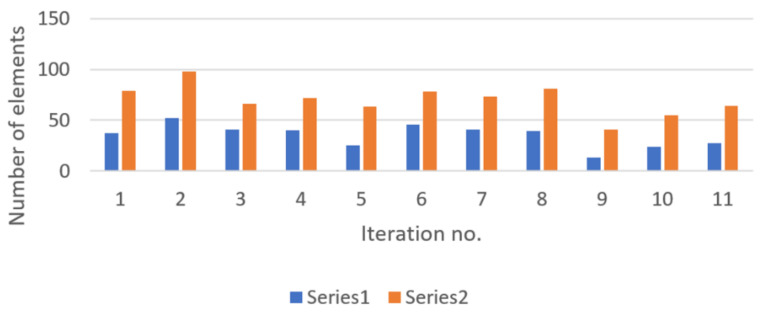
Detected devices (represented in blue—Series 1) versus real number of persons (represented in orange—Series 2) present in trains (traveling in tunnel).

**Figure 8 sensors-22-09025-f008:**
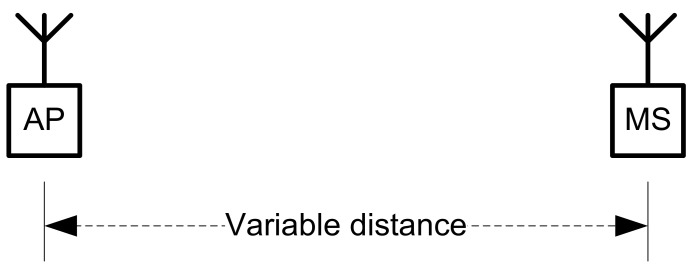
Setup for RX-based location tests.

**Figure 9 sensors-22-09025-f009:**
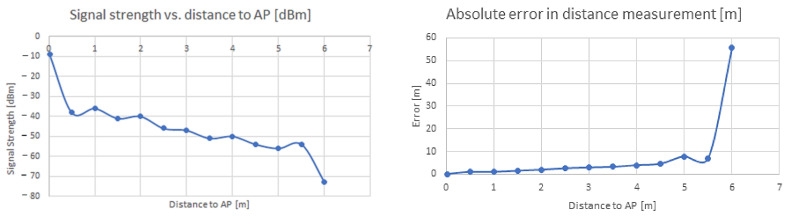
Variation of signal strength with distance (**left**), and absolute error in distance measurement (**right**)—equipment Huawei AP, mixed indoor environment (LoS + NloS).

**Figure 10 sensors-22-09025-f010:**
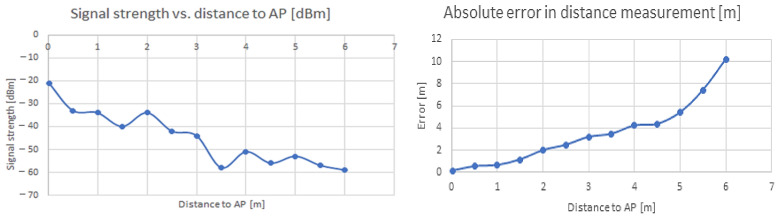
Variation in received signal strength with distance (**left**) and absolute error in distance measurements (**right**), Xiaomi AP, mixed indoor environment (LoS + NLoS).

**Figure 11 sensors-22-09025-f011:**
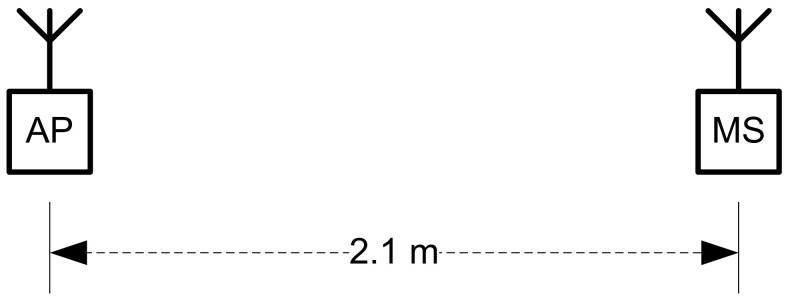
Test bed setup for signal strength variation measurement.

**Figure 12 sensors-22-09025-f012:**
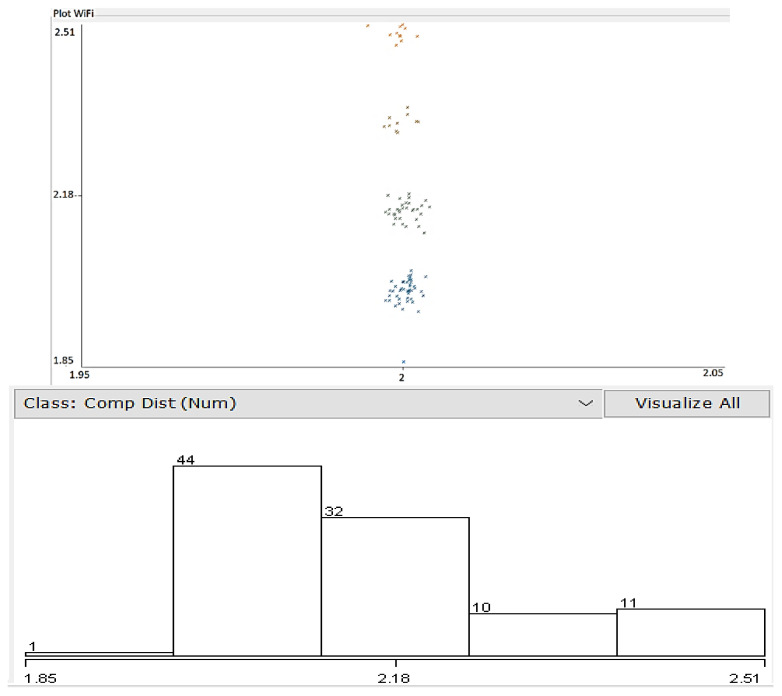
Distance to Wi-Fi AP computed from measured signal strength vs. real distance distribution, plotted in Weka environment. Colors represent belonging of distances to different clusters. Lower figure: Horizontal axis: computed distance, vertical axis: number of determinations.

**Figure 13 sensors-22-09025-f013:**
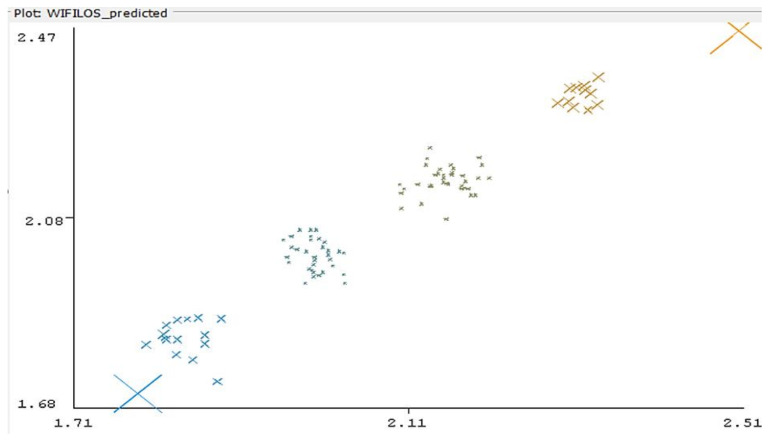
Weka Linear Regression classifier errors plotted for experimental results sequence 4 of tests (Vertical Axis—predicted values, Horizontal Axis—real distances). Colors represent belonging of errors to different clusters.

**Figure 14 sensors-22-09025-f014:**
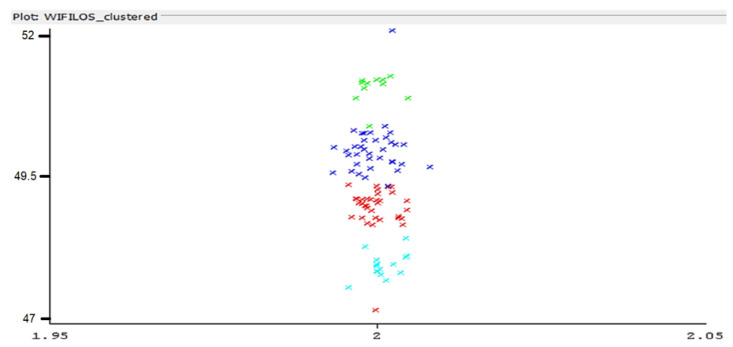
Expectation-Maximization clustering of recorded measurements—Wi-Fi constancy of received signal strength information, LoS condition—colors represent grouping of recorded values performed by the EM algorithm (horizontal axis: distance to AP, vertical axis: RSS level).

**Figure 15 sensors-22-09025-f015:**
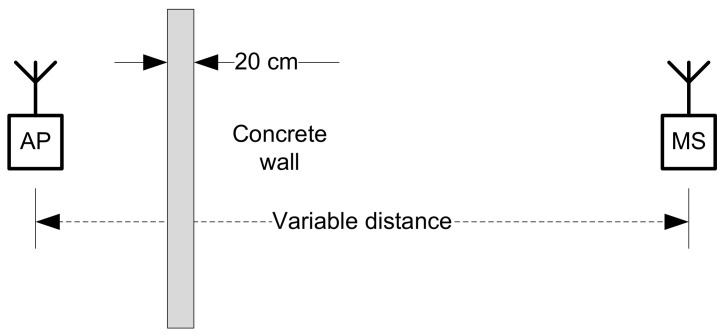
Test bed setup for BT signal strength indicator variation in stationary conditions—NLoS connection.

**Figure 16 sensors-22-09025-f016:**
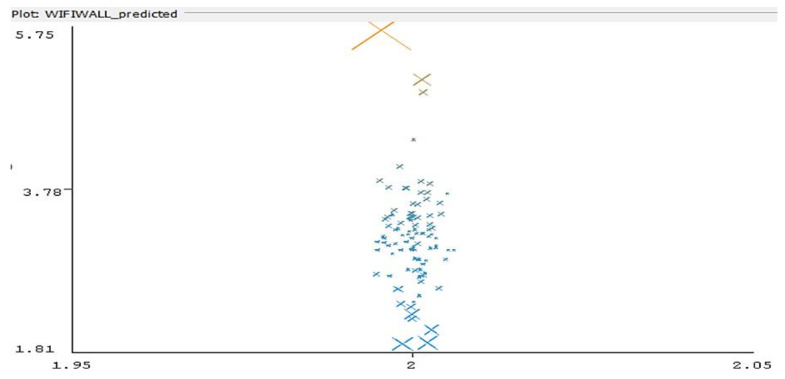
Error distribution in RSSI-based prediction of distance to AP, NLoS scenario—behind an armored concrete wall—colors represent belonging of errors to different clusters, as computed by linear regression algorithm (Vertical axis: predicted distance, horizontal axis: real distance).

**Figure 17 sensors-22-09025-f017:**
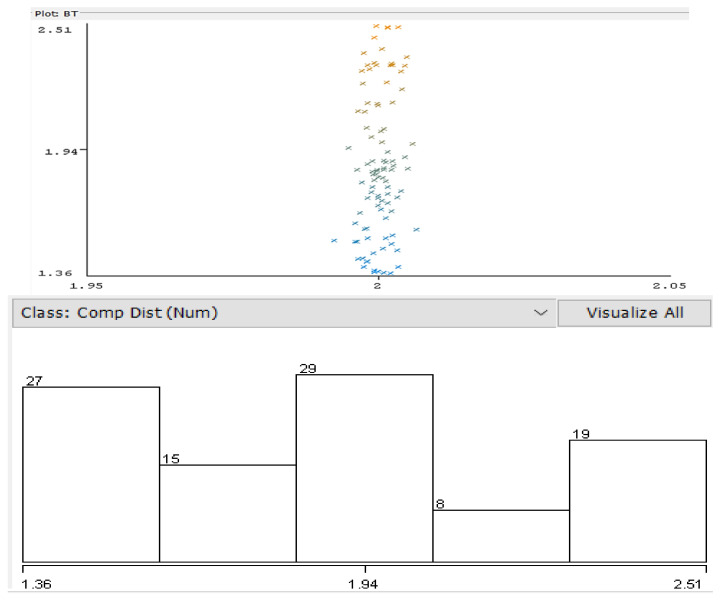
Distance to BT device computed from measured RSSI, vs. real distance distribution, plotted in Weka (horizontal axis: distance [m], vertical axis: number of events). The colors represent belonging to the different clusters, as classified by Weka with linear regression.

**Figure 18 sensors-22-09025-f018:**
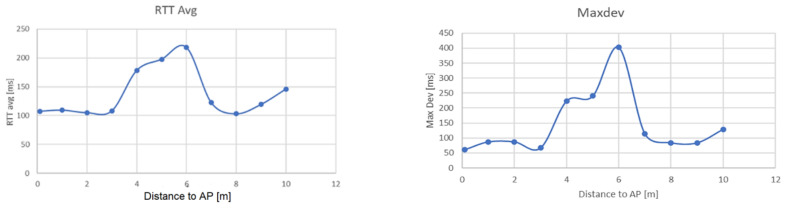
Variation of RTT (**left**) and Maximum deviation of recorded RTT (**right**) according to distance to AP, indoor, mixed path (0–8m LoS, 8–12m NLoS).

**Figure 19 sensors-22-09025-f019:**
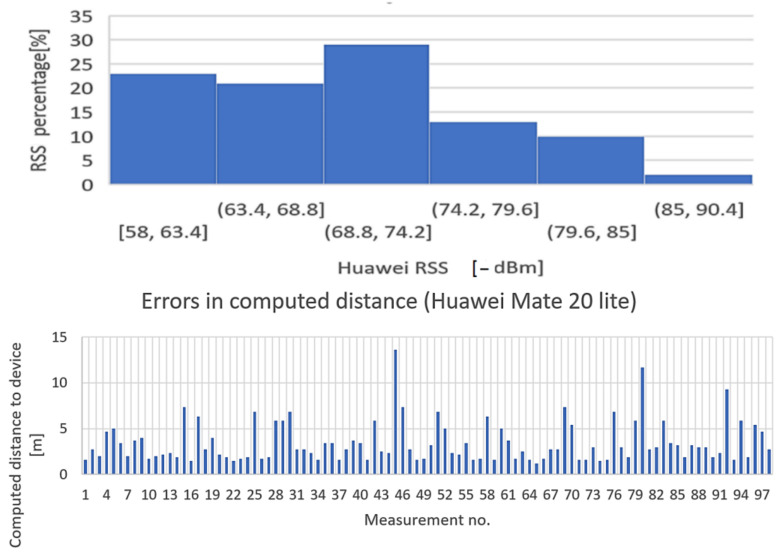
Variation in signal strength for fixed position of mobile station, LoS, d = 1 m. Equipment: Huawei Mate 20 Lite (Histogram + errors in computed distance).

**Figure 20 sensors-22-09025-f020:**
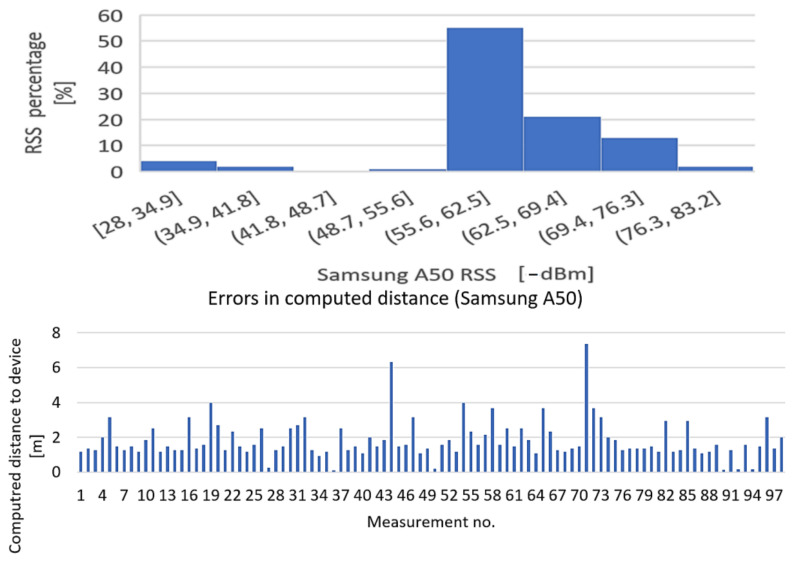
Variation in signal strength for fixed position of mobile station, LoS, d = 1 m. Equipment: Samsung Galaxy A50 (Histogram + errors in computed distance).

**Figure 21 sensors-22-09025-f021:**
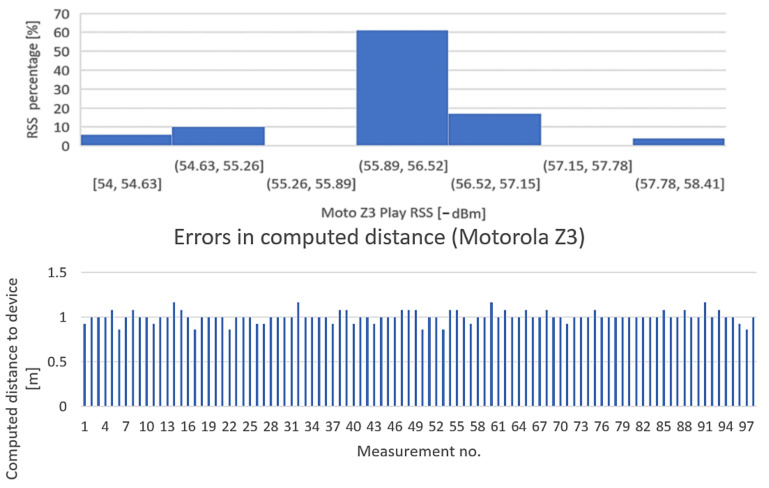
Variation in signal strength for fixed position of mobile station, LoS, d = 1 m. Equipment: Motorola Z3.

**Figure 22 sensors-22-09025-f022:**
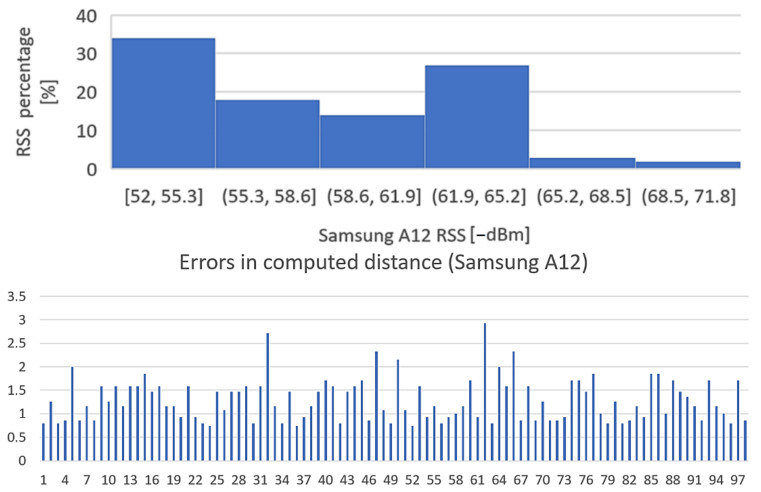
Variation in signal strength for fixed position of mobile station, LoS, d = 1m. Equipment: Samsung Galaxy A12.

**Figure 23 sensors-22-09025-f023:**
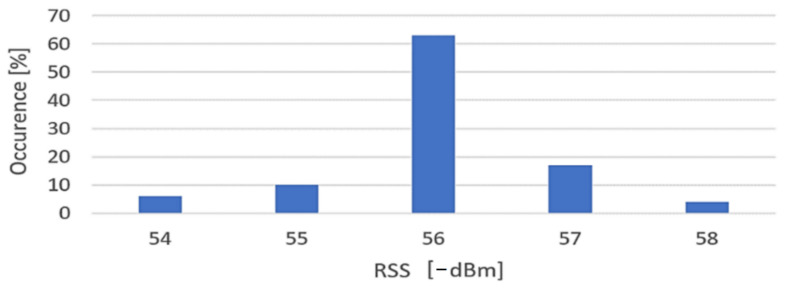
Regularized histogram for Motorola Z3, stationary conditions, indoors scenario, 1 m distance to transmitter, LoS.

**Figure 24 sensors-22-09025-f024:**
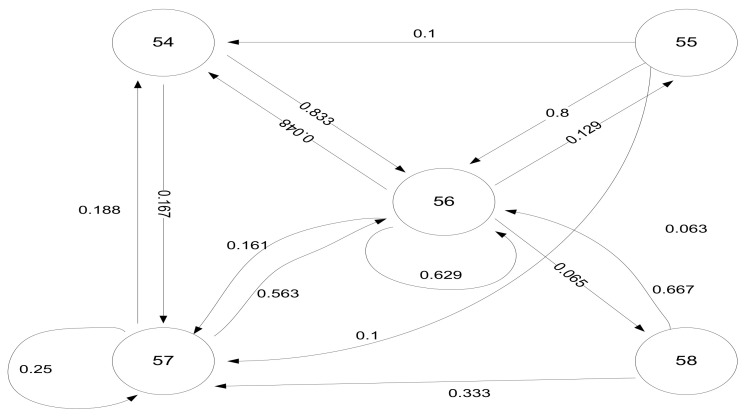
Transitions diagram (Markov Chains model—based on Motorola Z3 measurements, indoor conditions, LoS, 1 m distance to transmitter).

**Figure 25 sensors-22-09025-f025:**
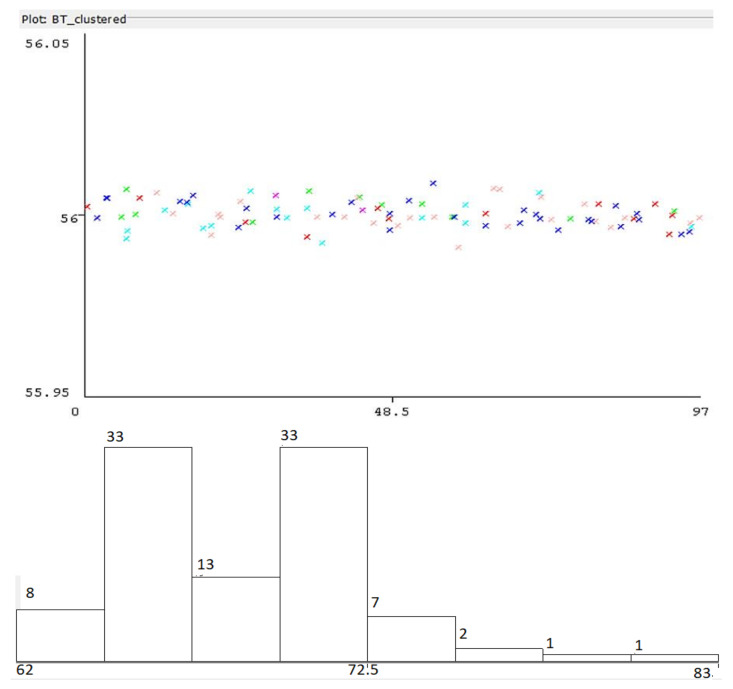
Dispersion of received signal strength recorded values during the test with BT source, LoS condition, 2 m from transmitter (horizontal axis—level [dBm], vertical axis: number of receptions). Colors in the above figure show belonging to time clusters when values were recorded.

**Figure 26 sensors-22-09025-f026:**
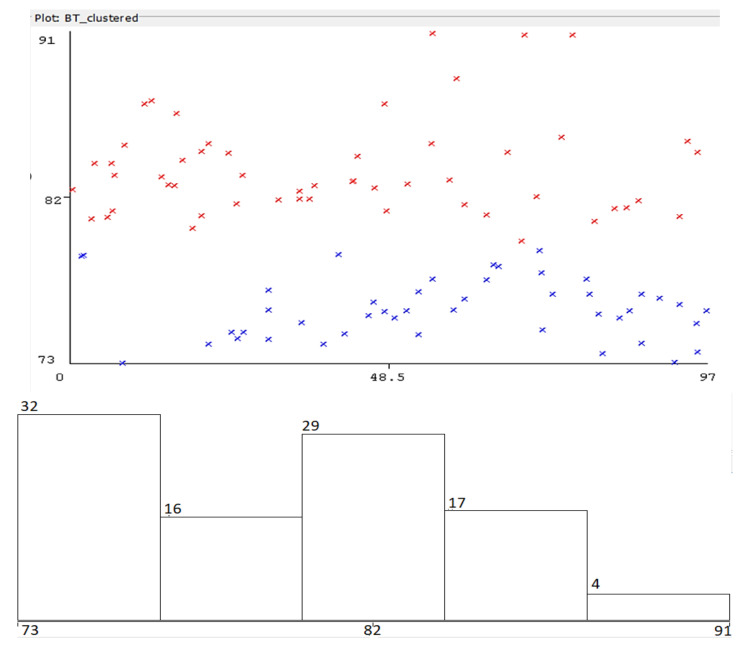
Dispersion of received signal strength recorded levels, with the BT source behind a wall of concrete (NLoS), 2 m distance (horizontal axis—level [dBm], vertical axis: number of receptions). The colors represent belonging to the different classified clusters.

**Table 1 sensors-22-09025-t001:** Position of Bluetooth technology amongst similar communication technologies.

Standard	Frequency [GHz]	Range Indoors [m]	Range Outdoors [m]	Transmission Power [mW]
Bluetooth	2.4–2.5	1–10	1–1000	−20 dBm (0.01 mW) to +20 dBm (100 mW)
BLE	2.4–2.5	1–10	1–100	10
ZigBee	2.4	20	1500	1
nRF	2.4–2.5	1–50	1–1000	1
IEEE 802.11 b/g/n (Wi-Fi)	2.4, 5	70	230	100 mW (20 dBm) on 2.4 GHz and 200 mW (23 dBm) on 5 GHz
LTE	Band 2: 1.9 Band 5: 0.85 Band 4: 1.7/2.1	Cell	Cell	Variable
5G (C-V2X)	FR1: <6 FR2: 24.25 to 71.0	Cell	Cell	Variable

**Table 2 sensors-22-09025-t002:** Values recorded in Subway Station 1.

BT Discovered Devices (Far Located—FL)	BT Discovered Devices (Near Located—NL *)	Counted Number of Persons in Station	Percentage of Discoverable Persons [%]
14	6	29	68.97
9	5	30	46.67
8	5	30	43.33
8	4	30	40.00
7	8	35	42.86
5	4	14	64.29
18	5	38	60.53
4	9	35	37.14
4	6	30	33.33
3	6	30	30.00
5	8	32	40.63

* Near located: all RX values better than −75 dBm, far located all RX values worse than −75 dBm. The average value for one hour of measurements taken every 5 min reached 46.15%.

**Table 3 sensors-22-09025-t003:** Values recorded in Subway Station 2.

BT Discovered Devices	Counted Number of Persons in Station	Percentage of Discoverable Persons [%]
25	80	31.25
14	60	23.33
18	65	27.69
2	4	50.00
5	20	25.00
6	20	30.00
7	25	28.00
12	30	40.00
14	41	34.15
17	50	34.00
7	23	30.44
16	63	25.39

**Table 4 sensors-22-09025-t004:** Recorded values of signal strength, Huawei AP, mixed indoor environment (LoS from 0 to 12 m + NLoS beyond 12 m).

Distance to AP [m]	Measured Distance to AP [m]	Signal Strength [−dBm]	Wi-Fi Speed [Mbps]	Observations
0.02	0.04	9	65	LoS
0.5	0.99	38	65	LoS
1	1	36	65	LoS
1.5	1.48	41	65	LoS
2	1.94	40	65	LoS
2.5	2.58	46	65	LoS
3	2.92	47	65	LoS
3.5	3.22	51	65	LoS
4	3.94	50	65	LoS
4.5	4.59	54	65	LoS
5	7.85	56	65	LoS
5.5	7	54	65	LoS
6	55.61	63	65	NLoS

**Table 5 sensors-22-09025-t005:** Recorded values of received signal strength, Xiaomi AP, mixed indoor environment (LoS from 0 to 12 m + NLoS beyond 12 m), Wi-FI channel 13, frequency 2472 MHz.

Distance to AP [m]	Measured Distance to AP [m]	Signal Strength [−dBm]	Wi-Fi Speed [Mbps]	Observations
0.02	0.14	21	86	LoS
0.5	0.54	33	86	LoS
1	0.65	34	86	LoS
1.5	1.15	40	86	LoS
2	2	44	86	LoS
2.5	2.48	42	86	LoS
3	3.2	44	86	LoS
3.5	3.48	58	86	LoS
4	4.24	51	86	LoS
4.5	4.38	56	86	LoS
5	5.43	53	86	LoS
5.5	7.45	57	86	LoS
6	10.24	59	86	NLoS

**Table 6 sensors-22-09025-t006:** Recorded values during RTT measurements.

Distance to AP [m]	Time [ms]	Avg. RTT [ms]	Maxdev RTT [ms]	Observations
1	58,999	107.545	61.204	LoS
2	59,917	109.593	87.345	LoS
3	59,074	105.076	87.154	LoS
4	59,972	108.297	66.909	LoS
5	58,948	178.239	223.895	LoS
6	59,920	197.794	240.098	LoS
7	59,912	218.362	402.759	LoS
8	59,839	122.5	114.177	NLoS
9	59,883	103.148	84.001	NLoS
10	60,079	119.444	83.826	NLoS
11	59,974	146.09	128.769	NLoS

## Data Availability

Not applicable.

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
