# Peer review of "An Experimental Assessment of People’s Location Efficiency Using Low-Energy Communications-Based Movement Tracking"

_sensors, 2022, doi:10.3390/s22229025_

Round 1
Reviewer 1 Report
Thank you for submitting the paper, here are some remarks.
1. The paper needs modification, it's closer to a report than a paper, and authors must clarify the nobility of their work.
2. All info stated in the paper should be cited unless they are well-known, examples of this:
the three RSS averaging methods, since the reader may be interested in the topic but can't find the reference.
Equations 1-3, also you need to clarify where (24, -51.3, and -49.2) came from.
citation are required for lines [433-436], [479-483], [701-704], [754-762] ... etc.
3. It's not clear in the text how devices are distributed through the experiments, a sketch of how devices are distributed will make things clear.
4. A table that compares BT with other wireless technologies is important.
5. Why did you use RSSI instead of RX although you mention that RSSI depends on the equipment used?
6. Many figures are required to be reproduced to be clear.
7. The paper is extended with no need, the reader wants details of the experiments but needs also the results to be comprehensive and clear, for example, Fig8 + Fig10 can be made on one figure, and the same thing applies to Fig9+ Fig11... etc. Also, you did not present the effect of power averaging to overcome the problem that occurred in Fig 9+11.
8. You need to make the experiment setup separate from the results; therefore, you need to dedicate a whole section for all your experiments' setups and then a detailed section for results for each experiment.
Author Response
Letter to Reviewer #1:
Many thanks for your time and willingness to review the article. The following are answers to your kind observations (R: Reviewer observation; A: Answer)
R: 1. The paper needs modification, it's closer to a report than a paper, and authors must clarify the nobility of their work.
A: The motivation of the research has been stated more clearly in the introduction section of the paper. The length has been shortened as much as possible, keeping important information and experimental results still available to the readers. Some of the figures were reconsidered and/or reconstructed to improve readability.
R: 2. All info stated in the paper should be cited unless they are well-known, examples of this: the three RSS averaging methods, since the reader may be interested in the topic but can't find the reference.
A: Three new citations have been introduced (on page 12, first new paragraph, citations [35] to [37])
R: Equations 1-3, also you need to clarify where (24, -51.3, and -49.2) came from.
A: Citation has been introduced in the main text before presenting the formulae.
R: citation are required for lines [433-436], [479-483], [701-704], [754-762] ... etc.
A: The text corresponding to pages [433-436] has been removed from the article. However, this information was initially retrieved from Bluetooth SIG, Inc. Headquarters, https://www.bluetooth.com/ .
To the text corresponding to pages [479-483] citation has been introduced.
The text corresponding to pages [701-704] in the original document has been modified and a citation has been introduced.
Text and figures between pages in the original document represent experimental results obtained by the author. The text has been reviewed and modified.
R: 3. It's not clear in the text how devices are distributed through the experiments, a sketch of how devices are distributed will make things clear.
A: Figures 1 and 2 have been replaced with diagrams showing placement of different equipment and personnel in the subway stations, and corresponding distances. For every other experiments, illustrative diagrams showing equipment have been introduced.
R:4. A table that compares BT with other wireless technologies is important.
A: Table 1 showing the required comparison has been included in the body text, introductory section.
R: 5. Why did you use RSSI instead of RX although you mention that RSSI depends on the equipment used?
A: Corrections have been made in the body text referring to experiments with signal strength.
R: 6. Many figures are required to be reproduced to be clear.
A: Figures 1 and 2 have been completely replaced with diagrams showing the test-bed setups. Figures 8,11, and 15 are new and show different setups for experiments performed in indoor environments. Figures 12, 17, 25, and 26 have been reconstructed for improved resolution (lower diagrams – upper diagrams are direct screenshots from Weka classifier and cannot be further processed). I would like to keep them because they give a more comprehensive, visual imaging of the dispersion of received signal values.
R: 7. The paper is extended with no need, the reader wants details of the experiments but needs also the results to be comprehensive and clear, for example, Fig8 + Fig10 can be made on one figure, and the same thing applies to Fig9+ Fig11... etc. Also, you did not present the effect of power averaging to overcome the problem that occurred in Fig 9+11.
A: Figures 8+10, 9+11 and similar others have been combined in a single figure, respectively. Experiments for averaging the power, related to figures 9-11 have not been performed yet, research is still in progress on this issue.
R: 8. You need to make the experiment setup separate from the results; therefore, you need to dedicate a whole section for all your experiments' setups and then a detailed section for results for each experiment.
A: Each experiment has been supplementary detailed by introducing diagrams with the setups of the test environment and placement of devices.
These answers have been also inserted in the attached file.

Reviewer 2 Report
The paper reports on experimental results on people tracking and localisation using Bluetooth and WiFi MAC scanners in a railway station. The results include statistical analysis as well as some interesting insights about challenges, etc. The paper is readable but not well written and structured. The presentation and formatting also need some improvement.
- Considering the popularity of the topic, the author should comment and explain on how the obtained results/statistics compare with prior published work on this subject, e.g. [1].
- The term 'near field - NF' in relation to Bluetooth should be reconsidered to avoid possible confusion with NFC. Similarly, the usage of term ToF as Round-trip-time is incorrect.
- The experimental setup, e.g. the layout of scanners, their configuration, distances, duration, etc needs to be explained in more detail.
Presentation/Formatting issues:
- At 31 pages, the paper is overly long and lacks focus. The author may consider shortening it by removing generic content, or by presenting only relevant information, e.g. a plot rather than an entire window screenshot.
- The paper needs to be proof-read for grammar mistakes
- Fig 3-4, the legend labels ('series1' and 'series2') could be more informative. The orange bars are described as brown in the caption. This should be checked for all other figures. The graphic resolution for some figures could be improved
- Fig 10, Sginal -> Signal
[1] Kurkcu, A., & Ozbay, K. (2017). Estimating Pedestrian Densities, Wait Times, and Flows with Wi-Fi and Bluetooth Sensors. Transportation Research Record, 2644(1), 72–82
Author Response
Answers for Reviewer 2 observations:
Many thanks for your thorough evaluation and observations! I hope that the below answers and actions required by your observations, will fulfill your requirements (R: reviewer request, A: answer):
R: The paper reports on experimental results on people tracking and localisation using Bluetooth and WiFi MAC scanners in a railway station. The results include statistical analysis as well as some interesting insights about challenges, etc. The paper is readable but not well written and structured. The presentation and formatting also need some improvement.
A: The paper has been reviewed and modified according to your requirements. It has been shortened and some phrases have been reformulated, new diagrams introduced, and several corrections on text have been performed.
R: Considering the popularity of the topic, the author should comment and explain on how the obtained results/statistics compare with prior published work on this subject, e.g. [1].
A: An explanation has been included in the main text, on page 2.
R: The term 'near field - NF' in relation to Bluetooth should be reconsidered to avoid possible confusion with NFC. Similarly, the usage of term ToF as Round-trip-time is incorrect.
A: The term “near field” has been reconsidered, and renamed as “close located”, referring to those detectable devices from which the RSSI levels were higher than -75 dBm, in contrast with “far located”, term used for denominating devices from which the received signal strength indicator was lower than -75 dBm. “ToF” has been removed from the body text, and the experiments only refer to RTT.
R: The experimental setup, e.g. the layout of scanners, their configuration, distances, duration, etc needs to be explained in more detail.
A: Explanations on the subway stations test bed setups have been introduced in Section 2, along with new diagrams in Figures 1 and 2, showing the positioning of sensors in the overall arrangement of the two stations (located in Bucharest Subway Network, namely Station Obor and Universitate, which are the among the busiest in the city).
R: Presentation/Formatting issues:
R: At 31 pages, the paper is overly long and lacks focus. The author may consider shortening it by removing generic content, or by presenting only relevant information, e.g. a plot rather than an entire window screenshot.
A: Overall, in different sections, several paragraphs and sub-sections have been reconsidered and/or removed, while trying to keep the understanding in the context. Most of the actions of removing text have been performed in the introductory part of the paper, as shown in Track Changes.
R: The paper needs to be proof-read for grammar mistakes
A: The paper has been entirely reviewed and grammar corrections performed where necessary.
R: Fig 3-4, the legend labels ('series1' and 'series2') could be more informative. The orange bars are described as brown in the caption. This should be checked for all other figures. The graphic resolution for some figures could be improved.
A: Explanations have been included for Series 1 and Series 2 in all figures. Probably the displayed color may depend on the monitor. The word “brown” in the explanation has been replaced by “orange”.
R: Fig 10, Sginal -> Signal
A: Could not find this reference.
These replies have been also inserted in the attached file.

Round 2
Reviewer 2 Report
The revised version addressed most but not all comments.
- Fig 10, the caption within the chart has a typo: Sginal -> Signal
- Section 3.7 heading still contains a term 'ToF' and needs to be changed.
- The text and some tables still contain a term 'Near field'
- The quality (resolution) of Figure 12 is quite low. There is no need to include a screenshot of the entire window, just the relevant information. This holds for all other figures showing the entire window.
- Given that MAC address tracking presents a serious privacy concern, the current android contain MAC address randomisation capability [1][2][3], making it much more difficult to track and count people. Similarly, Bluetooth Low Energy has an address randomisation privacy mechanisms [2][3]. The paper needs to discuss these factors and their potential impact on the results and the methodology in general.
- The quality/resolution of the chart could be improved
- Since the revised version has 'tracked changes' mode, it is quite difficult to see if the manuscript has been actually shortened.
[1] Torkamandi, P., Kärkkäinen, L., Ott, J. (2021). An Online Method for Estimating the Wireless Device Count via Privacy-Preserving Wi-Fi Fingerprinting. In: Hohlfeld, O., Lutu, A., Levin, D. (eds) Passive and Active Measurement. PAM 2021. Lecture Notes in Computer Science(), vol 12671. Springer
[2] V. Dyo and J. Ali, "Privacy-preserving Identity Broadcast for Contact Tracing Applications," 2021 Wireless Days (WD), 2021, pp. 1-6, doi: 10.1109/WD52248.2021.9508281.
[3] S. Akiyama, R. Morimoto and Y. Taniguchi, "A Study on Device Identification from BLE Advertising Packets with Randomized MAC Addresses," 2021 IEEE Int'l Conf. on Consumer Electronics-Asia (ICCE-Asia), 2021, pp. 1-4, doi: 10.1109/ICCE-Asia53811.2021.9641870.
Author Response
Thank you for the observations. Please find below the answers.
- Fig 10, the caption within the chart has a typo: Sginal -> Signal
Answer: Two figures having similar errors have been corrected: Figure 9 and Figure 10. Thank you for the observation.
- Section 3.7 heading still contains a term 'ToF' and needs to be changed.
Answer: This abbreviation has been deleted.
- The text and some tables still contain a term 'Near field'
Answer: Corrections have been made in:
- Table 2 heading, replacing the term Near field with Near located (NL)
- The quality (resolution) of Figure 12 is quite low. There is no need to include a screenshot of the entire window, just the relevant information. This holds for all other figures showing the entire window.
Answer: Sections from Figures 12, 13, 16, 17 (upper part), 25 (upper part), 26 (upper part) have been cut, some characters increased in dimension and sharpness improved.
- Given that MAC address tracking presents a serious privacy concern, the current android contain MAC address randomisation capability [1][2][3], making it much more difficult to track and count people. Similarly, Bluetooth Low Energy has an address randomisation privacy mechanisms [2][3]. The paper needs to discuss these factors and their potential impact on the results and the methodology in general.
[1] Torkamandi, P., Kärkkäinen, L., Ott, J. (2021). An Online Method for Estimating the Wireless Device Count via Privacy-Preserving Wi-Fi Fingerprinting. In: Hohlfeld, O., Lutu, A., Levin, D. (eds) Passive and Active Measurement. PAM 2021. Lecture Notes in Computer Science(), vol 12671. Springer
[2] V. Dyo and J. Ali, "Privacy-preserving Identity Broadcast for Contact Tracing Applications," 2021 Wireless Days (WD), 2021, pp. 1-6, doi: 10.1109/WD52248.2021.9508281.
[3] S. Akiyama, R. Morimoto and Y. Taniguchi, "A Study on Device Identification from BLE Advertising Packets with Randomized MAC Addresses," 2021 IEEE Int'l Conf. on Consumer Electronics-Asia (ICCE-Asia), 2021, pp. 1-4, doi: 10.1109/ICCE-Asia53811.2021.9641870.
Answer: a short comment in the Discussion section has been included, along with the recommended reference positions. The comment has been kept as short as possible, in order not to supplement the length of the article.
- The quality/resolution of the chart could be improved
Answer: All charts and figures in the paper have been increased in sharpness using Word Image Editor.
- Since the revised version has 'tracked changes' mode, it is quite difficult to see if the manuscript has been actually shortened.
Answer: The original length of the article was 31 pages. The revised version has been reduced to 27 pages. A non-tracking changes file has been attached for an easier reading.
